# Investigating Young Employee Stressors in Contemporary Society Based on User-Generated Contents

**DOI:** 10.3390/ijerph182413109

**Published:** 2021-12-12

**Authors:** Ning Wang, Can Wang, Limin Hou, Bing Fang

**Affiliations:** Department of Information Management, School of Management, Shanghai University, Shanghai 200444, China; ningwang@shu.edu.cn (N.W.); caseywang@shu.edu.cn (C.W.); liminhou@shu.edu.cn (L.H.)

**Keywords:** employee stressors, user-generated content, qualitative content analysis, JD-R theory, mental health

## Abstract

Understanding stressors is an effective measure to decrease employee stress and improve employee mental health. The extant literature mainly focuses on a singular stressor among various aspects of their work or life. In addition, the extant literature generally uses questionnaires or interviews to obtain data. Data obtained in such ways are often subjective and lack authenticity. We propose a novel machine–human hybrid approach to conduct qualitative content analysis of user-generated online content to explore the stressors of young employees in contemporary society. The user-generated online contents were collected from a famous Q&A platform in China and we adopted natural language processing and deep learning technology to discover knowledge. Our results identified three kinds of new stressors, that is, affection from leaders, affection from the social circle, and the gap between dream and reality. These new identified stressors were due to the lack of social security and regulation, frequent occurrences of social media fearmongering, and subjective cognitive bias, respectively. In light of our findings, we offer valuable practical insights and policy recommendations to relieve stress and improve mental health of young employees. The primary contributions of our work are two-fold, as follows. First, we propose a novel approach to explore the stressors of young employees in contemporary society, which is applicable not only in China, but also in other countries and regions. Second, we expand the scope of job demands-resources (JD-R) theory, which is an important framework for the classification of employee stressors.

## 1. Introduction

Stress is an integral part of employees’ lives and occurs in a wide variety of work environments [1]. Employee stress is frequently defined as a work or personal related feeling of difficulty, frustration, depression, or tension. It is deemed as a harmful part of the work environment [2]. First, it can seriously undermine employee well-being, thereby provoking health-related impairments globally [3]. Second, employee stress may affect an employee’s family or personal life [4]. Third, fueled by adverse psychosocial working conditions, employee stress increases employee absence [5]. According to the World Health Organization (WHO), more than 350 million working people worldwide suffer from depression, with an increase in about 18% in the past decade. The prevalence rate is higher among young people [6].

A good understanding of employee stressors is an effective measure to decrease employee stress. Therefore, a significant number of research on employee stressors across various occupations have been developed over the last forty years. According to [7], there are mainly two types of factors leading to employee stress: work-related stressors and family life-related stressors.

Work-related stressors can be divided into direct stressors and indirect stressors. In terms of direct stressors, performance stress is the top stressor for employees [8]. Performance stress is the urgency to improve performance in order to achieve desirable consequences and avoid negative consequences [9]. Employees who experience performance stress understand that meeting or exceeding performance expectations can lead to promotions, raises, and other benefits, while failing to meet expectations can lead to probation, termination, and other harmful outcomes. This experienced stress to improve performance affects employee well-being and stimulating psychological reactions. Second, the work environment is another direct stressor that can cause a threat to health and well-being. For instance, empirical studies demonstrate that a longer commute time is associated with lower job satisfaction and higher turnover tendency [10]. Apart from commute time, several studies have shown that poor working conditions (e.g., workplace ostracism and bullying) are associated with negative mental health effects such as depression and anxiety [11,12,13]. Moreover, unfair behavior in the working environment can also cause stress for employees [14,15,16]. Finally, resource shortages are another direct stressor. Firms strive to achieve more with less input in order to make more profit, satisfy growing customer demand, or respond to new opportunities in the marketplace. Employees will not have access to learning or exercising initiative [17]. There is little doubt that resource shortages lead to employees putting in more hours every year, with consequences ranging from burnout and fatigue to absenteeism, declining health, and strained interpersonal relationships [18,19,20].

Besides the direct stressors of work, there are another three indirect stressors. First, the negative effects of technology-related stressors (including social media addiction, prolonged working hours due to instant message, difficulties of operation of IT equipment, etc.) can cause stress by reducing employee productivity [21,22,23]. Second, emotional exhaustion is considered a key factor of job-induced stress, which is a stress response to the enormous emotional requirement dedicated to an organization to enhance organizational performance [17,24,25,26]. Third, employees can be “infected” with depression, anxiety, and stress-related illnesses when organizations hire employees from other unhealthy organizations (those with a high prevalence of mental health disorders) [27].

In addition to work-related stressors, studies have also identified family life as an important employee stressor. These studies focus on the following aspects: family financial stress, educational stress, and housing stress. First, employee stress is higher when they experience higher levels of family financial stress. Researchers suggest high rates of mental health issues among low-income families and their children [28,29,30,31]. Second, the considerable cost of education makes it impossible for employees to guarantee their children adequate education, which makes them feel overburdened and stressed [32]. Third, in recent years, pressing housing issues such as high housing costs, overcrowding, and job-housing imbalance have forced huge pressure on young employees, especially in developing countries such as China [33,34,35].

The above studies mainly focus on a singular stressor among various aspects of work and life such as performance stress, technology stress, commuting stress, etc. To our best knowledge, however, the literature has not presented a clear picture of all the comprehensive stressors from work and life that young employees are experiencing in contemporary society. In addition, there is research summarizing the previous research results on the stressors for young employees, but it does not verify these stressors using the real data [36]. Furthermore, researchers all propose hypotheses on a stressor beforehand subjectively, and then validate it based on observed data. In our study, we did not make any hypotheses on the kinds of stressors beforehand, rather, we focused on discovering the stressors directly from free narrations online and summarized the discovered stressors using the job demand-resource model (JD-R model). The JD-R model is a framework that is applied in the classification of employee stressors; it analyses the development process of any stressor following the demand process or resource process. Demand process refers to the job demands that lead to constant overtaxing and, in the end, to stress and exhaustion. Resource process refers to the lack of resources complicating the meeting of job demands, which also further leads to stress and withdrawal behavior [37]. With the JD-R model framework as shown in Figure 1, any stressor for employees can be categorized [38].

Another research gap is that the extant literature generally uses questionnaires or interviews to obtain data. These research methods are conventional and have the following two advantages: (1) they save time, money and manpower; and (2) the results are easier to quantify. However, the problem with such measures is that the same test object provides all the information and therefore the statistical relationship between the structures may be inflated due to common source bias [32]. In addition, the data obtained through questionnaires and interviews are subjective and lack authenticity, which may make the study results unrepresentative. For this reason, we systematically investigated the multiple stressors based on user-generated contents (UGC). UGC has emerged as a promising source of data to ascertain the actual thoughts and opinions of individuals [39]. UGC is easily accessible, with detailed information that can be processed efficiently and cost-effectively using advanced natural language processing (NLP) and machine learning technologies. As existing technologies still cannot fully grasp human language, a hybrid approach that combines the machine and human was applied in our research. Machine learning can efficiently extract keywords and sentences while human interventions aggregate their meanings.

In this paper, we introduced a machine–human hybrid approach to conduct qualitative content analysis of UGC to obtain comprehensive stressors. We divided the obtained stressors into mainly four categories: working process, family life, social media, and subjective cognitive bias. Finally, we give feasible suggestions to alleviate stressors.

There are two contributions in this paper. First, we systematically studied the stressors for young employees in contemporary society and identified some new stressors. Second, to obtain comprehensive and objective data, we developed a machine–human hybrid approach to gain insights into stressors for Chinese young employees.

The rest of this paper is organized as follows. Section 2 describes the study design and the methods applied for collecting data. Section 3 presents the empirical results derived from the UGC analysis. Section 4 discusses the implications for management practitioners. Section 5 concludes the paper with limitations and suggests future research directions.

## 2. Study Design

We developed a machine–human hybrid approach to conduct qualitative content analysis for the identification of stressors for Chinese young employees from UGC. The proposed approach can be divided into five stages (see Figure 2). First, we collected the raw UGC from a Chinese Q&A site. Second, we identified the informative sentences that can directly represent the stressors for Chinese young employees from the raw UGC. Third, using a language model named word2vec, we converted text into word embeddings that can be calculated to train word embeddings. Fourth, we clustered word embeddings and randomly sample words from each cluster in order to find the stressor-related keywords. Final, we extracted the original sentences of the keywords from the UGC, manually reviewed the chosen sentences, and analyzed the details of the stressors.

### 2.1. Stage 1: Data Collection

We collected the raw UGC from “Zhihu”, the largest Chinese social question and answering platform. Zhihu, which started their business in December 2010, provides long-term detailed answers and discussions raised by the users. As of December 2020, the total number of questions and answers on Zhihu had exceeded 44 million and 240 million, respectively. Users of Zhihu can openly and freely express their opinions on various topics. In addition, Zhihu’s data are available to the public and can be used for opinion mining [40,41].

According to the literature, we choose several keywords that could directly point out that young employees were under stress. We searched the questions containing these keywords on Zhihu, and crawled responses to the three questions: “What makes young people feel tired?”, “Why do so many employees work overtime in China? and what are they doing?”, and “Why do young people in China have no desires?”. Here, “burn out”, “work overtime”, and “desirelessness” can all infer people are under stress [17,37,42]. There are many responses to the three questions, which contain rich detailed information about the things young employees have suffered, and the stress on young employees from the things suffered. There were 3336 responses (a total of 25,044 sentences) to the first question, 2211 responses (a total of 16,589 sentences) to the second question, and 7073 responses (a total of 49,955 sentences) to the third question.

### 2.2. Stage 2: Identify Informative Content

UGC contains a substantial amount of sentences that do not relate to the stressors. To improve the accuracy of subsequent machine-learning, we identified the informative sentences that can directly express stressors for employees. We treated the identification process as a classification task: the classifier will identify the informative and noninformative sentences, and label them as 1 and 0, respectively. The Convolutional Neural Network (CNN) model was chosen to train the classifier. As an important basic model in deep learning, CNN can automatically extract key features and has a short training time to avoid the tedious process of manual feature extraction [43,44]. In addition, large-scale network implementation is much easier with CNN than with other neural networks [44]. The process of classification can be divided into three steps. In step 1, we manually labeled a small set of sentences as informative/noninformative sentences to build a set of training data. We randomly selected 1000 sentences from the answers of each question, with a total of 3000 sentences. We manually judged whether each sentence represented the stressors for young employees: if a sentence represents the stressors for young employees, it is labeled as 1; otherwise it will be labeled as 0. In step 2, we trained the text classifier. Since texts cannot be directly calculated, we mapped the raw UGC data onto sentence embeddings using Bidirectional Encoder Representation from Transformers (BERT) model. With the architecture of multi-layer bidirectional transformer encoder, BERT is empirically powerful in many natural language processing (NLP) tasks. Unlike other traditional language models such as RNN, it is an unsupervised model and does not require manual intervention [45]. The sentence embeddings are input in the CNN model to train the classifier. In step 3, the trained classifier automatically identifies informative sentences from the UGC.

### 2.3. Stage 3: Preprocess UGC and Train Word Embeddings

As mentioned earlier, texts cannot be directly calculated, so we needed to preprocess the informative sentences and map the words onto word embeddings. The process of preprocessing can be divided into two steps. In step 1, we segmented the Chinese sentences. Since the sentences were written in Chinese, it was necessary to split the sentences into words. We used “Jieba”, a Python software for Chinese text segmentation, to segment the Chinese sentences into a corpus. It is worth noting that, as mentioned earlier, we collected responses from three questions. We segmented the responses to the same question into one corpus. In this way, we constructed three corpora. In step 2, we removed the stop words. In the corpus, there were many stop words (e.g., “but” and “and”) that would significantly affect the efficiency of NLP and need to be deleted. Table 1 illustrates the comparison between the raw and processed data.

We mapped the preprocessed words onto numerical word embeddings using the skip-gram model. The skip-gram model is a kind of language model, which is based on the artificial neural network. It can process large-scale textual data and capture the contextual information of the text [46]. We implemented the skip-gram model through the Word2Vec package in Python. The training results were saved as word embeddings for use during the subsequent stages.

### 2.4. Stage 4: Cluster Key Words

To efficiently review the informative sentences, we clustered the words in the corpus into several groups based on word embeddings. We applied the “k-means” algorithm to cluster the keywords. This algorithm was implemented with the following steps. First, several words were randomly designated as the cluster centers of each group. Second, the distance between words and each cluster center were measured. Third, the words were classified into the nearest cluster. Fourth, each cluster center was recalculated. The iterations of the second and fourth steps continued until there was no further change in each cluster center. We clustered the words into 30 groups for each question, with a total of 90 groups. The words in the same group usually had similar semantics. The clustering results were saved for use in the next stage.

### 2.5. Stage 5: Manually Extract the Details of Stressors for Young Employees

Since the existing NLP technology is not able to really understand the semantics of the text, we have to manually analyze the meaning of the sentence. We randomly selected four sample words from each group, and extracted the sentences that contained the sample words from the UGC. We manually reviewed the extracted sentences and identified the details of stressors for young employees. Two professors and two graduate students participated in this operation. They conducted an evaluation based on the four eyes principle, which can effectively reduce serious error in judgment [47]. Initially, they reviewed the sentences separately, and then verified the consistency. If no agreement was reached, they continued their discussion until agreement was reached.

## 3. Results

After reviewing the chosen sentences, we identified four main categories of stressors for Chinese young employees mentioned frequently by users of “Zhihu”: working process, family life, social media, and subjective cognitive bias. Figure 3 illustrates a holistic view with detailed information emerging from the data.

The dark blue central node represents employee stress. The light blue nodes linked with the central dark node represent four categories. The keywords included in the four categories are listed nearby. The number in parentheses express the counts of responses that mention the keywords. Table 2 show the detailed information on each category, its keywords, and related original example sentences.

### 3.1. Working Process

The first was working long hours. There were four main keywords related to working long hours: “working overtime”, which can stress the body as well as the mind; “health” and “rest”, where there is talk that working long hours will cause health problems and reduce the amount of rest time; and “stay up late”, where people mentioned that they felt a lack of entertainment due to their working long hours, so had fun at night, sacrificing their rest time. The second subcategory was dead-end job. There were four main identified keywords related to dead-end jobs: “hope”, which is at the top of the keywords mined from the answers, where many people mentioned that even if they worked very hard, there was little hope in their jobs; “graduation”, where new employees complained that they were always confronted with job stress; “COVID-19 epidemic” and “competition”, where due to the outbreak of COVID-19, many people felt higher competition pressure. The third subcategory was a rough time during work. There were mainly four keywords related to this subcategory: “supervisor” and “push”, where many people complained that they were often pushed to work overtime by their supervisors; “reward“, where many people complained that they were not rewarded enough for what they produced; and “resign”, where many people said that they dare not resign even if the job was painful. The fourth subcategory was the social life at the workplace. There were two main keywords related to this subcategory: “colleagues” and “boss”, where many people said that they did not get along with their colleagues and bosses. The fifth subcategory was a dislike of their jobs. There were two main keywords related to this subcategory: “dream” and “love”, where many people said that they did not love their jobs and they chose to suppress their inner desires and dream. The sixth subcategory was the commute to work. There were three main keywords related to this subcategory: “stop”, ”bus”, and “commute”, where many people complained that they struggle through the long commutes. The seventh subcategory was a lack of social security. There were two main keywords related to this subcategory: “squeeze” and “exploit”, where some people said that they faced exploitation and pressure caused by their jobs due to the lack of social security. 

### 3.2. Family Life

Keywords emerged for family life were categorized under nine subcategories, see Table 3.

The first was low salary. There were two main keywords related to this subcategory: “salary” and “income”, where many people complained about the low salary. The second subcategory was high cost of family life. There were three main keywords related to this subcategory: “feed”, “marriage”, and “brother”, where many people said that they had to spend a considerable sum of money supporting their families. The third subcategory was high housing costs. There were three main keywords related to this subcategory: “buying houses”, “can’t afford”, and “house prices”, where a great number of people complained that the house prices were too high. Housing price was mentioned repeatedly in 5357 responses. The fourth subcategory was the high cost of getting married. There were two main keywords related to this subcategory: “get married” and “betrothal gifts”, where many people complained that getting married in China will cost too much. The fifth subcategory was daily consumption. There were two main keywords relating to this subcategory: “car loans” and “consumption”, where many people complained that daily consumption was too high to afford. The sixth subcategory was educational cost. There were three main keywords related to this subcategory: “kindergarten”, “kids”, and “tuition fees”, where many people complained that the educational cost was also too high to afford. The seventh subcategory was supporting aging parents. There were two main keywords related to this subcategory: “supporting aging parents” and “the elderly”, where many people mentioned that they had to spend a lot of money to support their aging parents. The eighth subcategory was health care expenses. There were two main keywords related to this subcategory: “health care” and” doctors”, where many people said that drugs, treatments, and seeing a doctor were very expensive, and they even dreaded being ill. The ninth subcategory was social pressure. There were two main keywords related to this subcategory: “parents”, where many people complained that they were under parental pressure to get married to anyone and “social situation”, where some people said that they had significant anxiety and depression due to poor ability to communicate in social situations.

### 3.3. Social Media

Keywords that emerged for social media can be categorized under three subcategories, see Table 4.

The first is the consumption of time and energy caused by social media. There were three main keywords related to this subcategory: “mobile phone”, where many people said that they spent a lot of time every day with their mobile phones unconsciously; and “video” and “Tik Tok”, where many people said that they watched videos on social media every night before bedtime, even when they came home late. The second subcategory was desire for materials aroused by social media. There were two main keywords related to this subcategory: “actual life”, where many people said that they dreamt of the high-end lifestyle spreading on social media, but were unable to make it happen in actual life, which would drag them into a deep depression; and “friend circle”, where many people said that they envied the lives of friends seen on social media. The third subcategory was lives occupied by busy working. There were three main keywords related to this subcategory: “punch a time clock”, “WeChat”, and “message”, where many people complained that since social media made it easier to contact them after work hours, their lives were further occupied by busy work.

### 3.4. Subjective Cognitive Bias

Keywords emerged for subjective cognitive bias were categorized under three subcategories, see Table 5.

The first subcategory was insatiable desire. There were three main keywords related to this subcategory: “desire”, “satisfy”, and “consumerism”, where many people said that they were passionate about the products out of their affordability. If their desires are not satisfied, they will feel frustrated. The second subcategory was the gap between the rich and poor. There were two main keywords related to this subcategory: “the gap between the rich and poor” and “the poor”, where many people said that through the information spreading on social media, they realize that there exists a growing gap that separates them from the richest people, and they are increasingly dissatisfied with their current situation. The third subcategory was the gap between dream and reality. There were two main keywords related to this subcategory: “gap” and “frustration”, where many people said that there exists a huge gap between their dreams and real life, which will lead to a sense of failure.

Based on the above results, we found that working process was usually at the top of the list of stressors for young Chinese employees. These have centered on fears that their jobs have no future: the chance of being promoted is low, and it is even possible to get fired. Long working hours and heavy tasks also make them feel tired and under great strain. On the other hand, high living expenses such as buying houses, supporting aging parents, and so on, weigh heavily on their minds. In addition, it is worth noting that besides external pressures, many young employees are also constantly under the stresses resulting from the gap between their dreams and real lives. People have an easy access to luxurious lifestyles through social media, which can produce the illusion that they are inferior and lead to people’s discontentment with their current situations.

The above results also provide further support for the JD-R theory. The identified stressors for young Chinese employees correspond precisely to the job demands, job resources, and personal resources. Job demands refer to those physical, social, or organizational aspects of the job that require sustained physical or mental effort and are therefore associated with certain physiological and psychological costs (e.g., exhaustion) [37]. The identified keywords related to “working long hours” and “lives occupied by busy working” express that these jobs require a lot of people’s time and energy. Job resources refer to those physical, psychological, social, or organizational aspects of the job that may do any of the following: (a) be functional in achieving work goals; (b) reduce job demands at the associated physiological and psychological costs; and (c) stimulate personal growth and development [37]. The identified sentences related to “dead-end job”, “social life at workplace”, “dislike their jobs”, “commute to work”, and “the consumption of time and energy caused by social media” all express that employees feel very tired of work due to their lack of job resources such as job skills, social skills, and so on, which can create strain on employees. Meanwhile, all of the identified sentences related to “family life” express that employees have to pay a considerable sum of money to raise their families, thus they have to work flat out to earn enough income from work. Personal resources refer to the beliefs people hold regarding how much control they have over their environment [38]. The identified keywords related to “desire for material aroused by social media”, “insatiable desire”, “the gap between the rich and poor”, and “the gap between dream and reality” only express that employees feel extremely miserable about the gap between their dreams and real lives. Additionally, our results also suggest that there are other stressors for employees that cannot be explained by JD-R theory: the sentences related to “rough time during the work” and “lack of social security” express that employees are often squeezed by their supervisor due to the lack of social guarantee of employee rights, which makes them feel exhausted. Table 6 compares our identified stressors with stressors used in the literature.

## 4. Discussion

Through qualitative content analysis conducted by a machine–human hybrid approach, we developed a comprehensive understanding of stressors for young Chinese employees. Our results matched the JD-R model exactly. In particular, we systematically integrated employee stressors and identified three kinds of new stressors that have not been mentioned in the previous literature.

First, we found that being squeezed by supervisors, as mentioned in Table 2, is a new kind of employee stressor that is related to employee well-being. To the best of our knowledge, this is a stressor that has not been mentioned in the previous studies. Abusive behavior of supervisors has been identified as a stressor for young employees that cannot be neglected [48]. With gradual improvement in the legal system, the abusive behavior of supervisors has gradually disappeared. However, in fact, in many enterprises, being squeezed by supervisors is still an inescapable stressor for young employees. Many supervisors are constantly putting pressure on the employees to increase labor output. This leads to excessive physical and mental output that puts tremendous pressure on employees in the environment.

Second, we found that affection from others (see Table 3 and Table 4) could also put a strain on employees. Previous research in organizations has demonstrated that the hard work of co-workers is for the affections from others, which are related to the employees’ psychological stress [49]. However, importantly, we found clear evidence that employees were influenced not only by colleagues, but also by social media and loved ones. Our results indicate that social media is one of the most important ways through which employees can be affected by others. In recent years, the use of social media has become a global phenomenon. Over two-thirds of Internet users have active accounts on social networking sites [50]. This leads to the frequent occurrence of social media fearmongering, which causes individuals to indulge in frequent negative comparisons between themselves and others, resulting in many unrealistic material desires. In addition, we also found that pressure from parents such as pushing for marriage could also put psychological strain on employees.

Finally, we found that subjective cognitive bias (see Table 5) plays an important role in causing employee stress; the ability to cope with such subjects cognitively can be seen as a kind of personal resource. In this respect, we extended the research results of the JD-R model. Most of the previous research on personal resources in the JD-R model emphasize the people’s perceptions of their control over their environment [38]. That is, it emphasizes the employees’ subjective feelings about whether they can finish the work on schedule. However, through data analysis, we found that personal resources should also be extended to one’s subjective feelings about the social environment. Due to subjective cognitive bias, employees feel that their life is worse than others, that there is a gap between dream and reality, and between material desire and reality. The frequent occurrences of social media fearmongering only aggravate these subjective cognitive biases. When these problems are not solved, they develop anxiety.

In light of our findings, several strategic directions have emerged to reduce employee stress and improve the employees’ physical and mental health.

First, we need to focus on the two kinds of stressors of being squeezed by supervisors and a dead-end job (Table 2) as they account for a large percentage of our results. Inadequate mechanism for employee protection is responsible for most of these stressors. Consequently, there is an urgent need to improve forms of interactions between employees and employers in the system of social and labor relations and develop social partnerships. Socially oriented employers should create favorable working conditions, improve the motivation and incentive systems, and ensure career growth [51]. In addition, other measures including social support, institutional guarantees, and legal protections should also be carried out to relieve employee stress. 

Second, the negative impact of pressure from others and subjective cognitive bias are serious concerns. Strategies for limiting such negative impacts are therefore necessary. Employees should be guided to establish clear self-cognition, correct values and beliefs, and judgement ability for Internet information. Besides, websites should also be able to automatically identify, filter, and remove misinformation. Such methods are an effective means of alleviating employee psychological stress and promoting their psychological health. 

Third, efforts should be made to reduce the mental stress, mental fatigue, and exhaustion experienced by employees with a family life. Suggested policy recommendations include improving social welfare protections for wage-earners. The provision of practical support for raising children, supporting the elderly, and buying houses could help employees to focus on their work more easily. In general, interactive mechanisms between employment and social security help to vigorously ensure and improve people’s well-being, maintain social stability, and promote social harmony and progress.

## 5. Conclusions

In conclusion, through the technology of natural language processing and machine learning to mine UGC on Zhihu, we proposed a novel approach to investigate the details on stressors for young Chinese employees. The proposed approach for stressor identification is applicable not only in China, but also in other countries and regions. We also contribute to JD-R theory by expanding the scope of resources (i.e., mechanism for protecting employee rights and personal cognition). In light of our findings, we offer valuable practical insights and policy recommendations to relieve stresses and improve the physical and mental health of employees. 

Our study can be extended in several directions. First, this research lies in the fact that our conclusions are based on the answers to the three questions from one social question and answer platform. The UGC data are from a single source, and do not contain socio-demographic and socio-economic characteristics of the users or the specific professions. In the future, we can utilize UGC data from various sources and conduct research together with other traditional social science methods. Second, our approach can be further improved, since it still needs human reading sentences for categorization. In the future, we can utilize automatic text classification techniques and text generation techniques for categorization and interpretation. Third, this research only discussed the stressors from the lack of resources and the increase in job demand. We encourage future research to further explore the factors in the context of the entire JD-R model and examine the direct and indirect effects on and of resources on the stressors. Finally, all of our samples were collected from China. We encourage future research to further explore the employee stressors in different countries and compare the differences.

## Figures and Tables

**Figure 1 ijerph-18-13109-f001:**
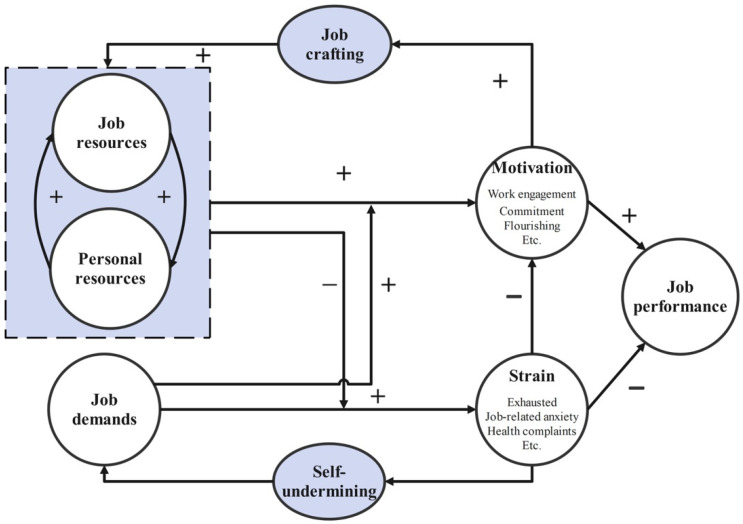
The job demand-resource model.

**Figure 2 ijerph-18-13109-f002:**
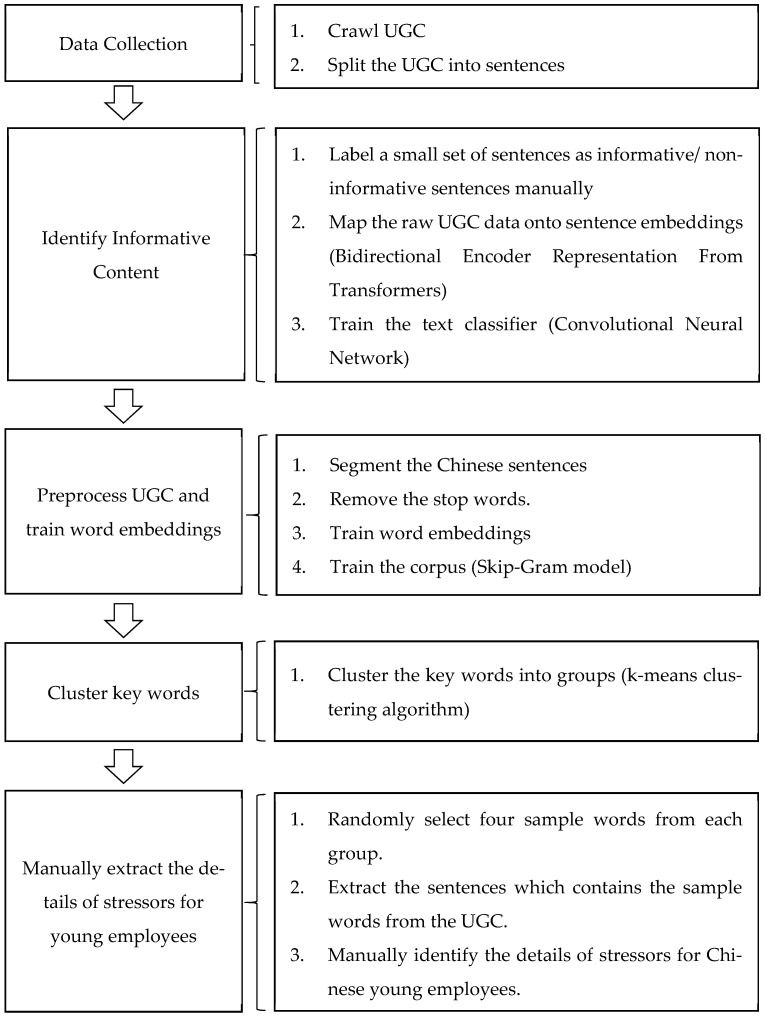
System architecture for identifying the stressors for Chinese employees from UGC.

**Figure 3 ijerph-18-13109-f003:**
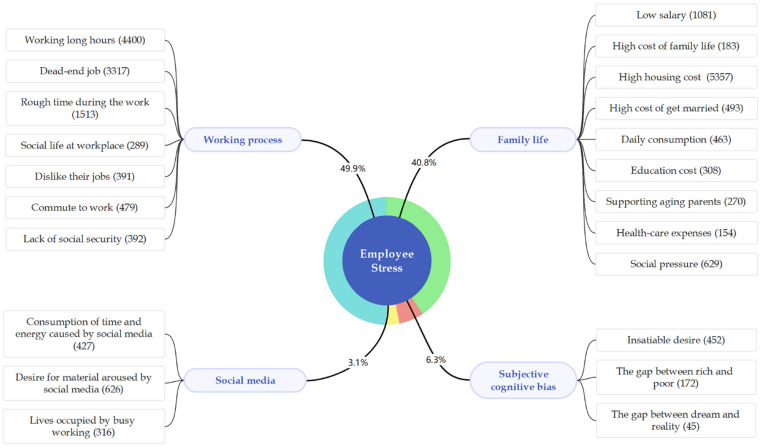
In-depth analysis of stressors for young employee in contemporary society.

**Table 1 ijerph-18-13109-t001:** Examples of the raw and processed data.

Raw Data	Processed Data
As a freshman, I have no experience at all, and every time I fail to do well, I will be considered incapable.	Freshman, experience, every time, fail, to do well, incapable
We no longer discuss houses, just the current housing price. This is a desperate, meaningless sad topic.	No longer, discuss, house, current housing price, desperate, meaningless, sad topic
Under continuous high-intensity and high-pressure work, the nerves are already very fragile, and the body is also suffering from major and minor problems.	Continuous, high-intensity, high-pressure work, nerves, fragile, body, suffering, major, minor, problems

**Table 2 ijerph-18-13109-t002:** Keywords indicating working process and examples of sentences.

Sub-Category	Keywords	Count	Examples of Sentences
Working long hours	Working overtime	338	I have to work overtime in the initial days.
			My overtime job makes me so exhausted.
	Health	337	I have to work around the clock every day, which will damage my health sooner or later.
			I often stay up all night for work, which leads to some trouble with my heart health.
	Rest	330	I dare not even ask for any vacation time to rest.
			I just get no rest, which has caused the mental strain oversized.
	Stay up late	142	I often stay up late, since I only have free time in the evenings.
			I often stay up late to play with my mobile phone.
	…	…	…
Dead-end job	Hope	717	I see no hope for the future, which is my worst hard ship.
			I see no hope for the future: the wage of my job is too low, and I often work over time.
	Graduation	372	I just graduated as a postgraduate. Now I am working in an Internet company, and I feel tired and stressful every day.
			As a fresh graduate, I am working from early morning till late night every day.
	COVID-19 epidemic	165	COVID-19 epidemic is affecting nearly all businesses.
			Unexpected COVID-19 epidemic has caused the increase in unemployment.
	Competition	116	With the economic downturn, the competition for new jobs or promotions is intense.
			In a highly competitive society, I have to work overtime to keep the job.
	…	…	…
Rough time during the work	Supervisor	172	Most of the time, my supervisors make unctuous promises.
			I worked hard for one year, but my supervisor takes all the credit.
	Push	162	My supervisors often push us to work overtime.
			Most of us are not voluntarily working overtime, we are pushed into work by supervisors.
	Reward	105	I work hard every day with little reward.
			The rewards I get are not proportional to the effort that I have paid.
	Resign	73	I dare not resign even if I am not happy at work.
			I dare not resign casually.
	…	…	…
Social life at workplace	Colleagues	155	I don’t want to cater to my supervisors, colleagues, and friends.
			There are often subtext in messages from my colleagues.
	Boss	134	I always disagree with my boss’s opinion, hovering on the verge of resignation and being fired.
			My boss tries to hold me down.
	…	…	…
Dislike their jobs	Dream	348	Only the rich-second-generation have the opportunities of realizing their dreams.
			I don’t have the material foundation to realize my dream.
	Love	43	I cannot find a job that I love, and my hobby is my real love.
			I really love taking photographs, but actually my job is to fix cars.
	…	…	…
Commute to work	Stop	160	The subway is so crowded in the morning
			Since I need to take the subway before 7:25, I have to buy my breakfast and walk to the stop in ten minutes.
	Bus	78	I have to wash up and eat breakfast quickly. Then I need to take the crowd bus or subway every morning.
	Commute	53	Office workers often have to spend more than two hours in commutes.
			It is too late for me to get home every day, since I have to spend an hour in commutes every day.
	…	…	…
Lack of social security	Squeeze	84	Supervisors squeeze everyone to work overtime since they need profits to improve an enterprise.
			Supervisors squeeze employees to work overtime, especially in declining businesses.
	Exploit	76	I have to keep working overtime, since the boss constantly exploits us.
	…	…	…

**Table 3 ijerph-18-13109-t003:** Keywords indicating family life and examples of sentences.

Sub-Category	Keywords	Count	Examples of Sentences
Low salary	Salary	366	Salaries are so low.
			I thought I could get a high salary if I worked hard, but actually I work hard with a low salary.
	Income	173	I have a very limited income.
			Although I work and study hard, my income is lagging far behind.
	…	…	…
High cost of family life	Feed	102	It’s hard to feed myself, let alone my family.
			No one will give me living expenses in vain to feed my whole family.
	Marriage	39	In my opinion, marriage is a waste of money.
			Maintaining marriage needs money.
	Brother	24	My parents and brother want to squeeze me, although their income is more than mine.
			I have to feed my brother and my family.
	…	…	…
High housing costs	Buying houses	675	I cannot afford to buy a house by myself in Guangzhou, where the housing prices are the lowest in the first-tier cities.
			I strive to save money to buy a house.
			It is impossible for me to buy a house and a car with such a low salary.
	Can’t afford	309	I can’t afford such expensive houses.
			I can’t afford a house and no one wants to marry me.
	House prices	234	Take Jinan’s house prices as an example, it has increased ten-fold in 20 years.
			The housing prices are desperate.
			Due to the high house prices, I can just rent a shabby house rather than buying a house.
	…	…	…
High cost to get married	Get married	330	I am so poor that I can’t get married or feed a child.
			In my opinion, getting married will cost too much money.
	Betrothal gifts	147	The expensive betrothal gifts make us give up to get married.
			Betrothal gifts are too expensive.
	…	…	…
Daily consumption	Car loans	138	I am overwhelmed by mortgage and car loans.
			It is difficult for us to pay the mortgage and the car loans, and I even want to sell my car.
	Consumption	104	With the soaring prices, it is difficult for families of ordinary consumption level to bear.
			Consumption levels have exceeded the income levels of ordinary families.
	…	…	…
Education cost	Kindergarten	96	In my small city, a kindergarten costs 3800 in a quarter.
			My children have to take part in the “specialty training class” since they are in kindergarten.
	Kids	93	I need to earn money to support my family and make my kids live a better life.
			We must take care of our parents, wives, and kids.
	Tuition fees	35	The mortgage, milk powder money, and tuition fees are all overwhelming me.
			I have to work hard to pay for my child’s tuition fee.
	…	…	…
Supporting aging parents	Supporting aging parents	136	I need to support my aging parents.
			Since my parents are growing old, I have to support my aging parents.
	The elderly	64	I have to earn money to support the elderly and children.
			I have to take care of four elderly people including my parents and my wife’s parents.
	…	…	…
Health-care expenses	Health care	67	With my parents growing old, I have to worry about their health care costs.
			I have a lot of pressure in health care, since the social security system is limited.
	Doctor	61	I wish I could afford the doctor’s bill when my parents are very ill.
			My mom has cancer. The doctor says that it can be cured, but we have to pay a large sum of money.
	…	…	…
Social pressure	Parents	400	When I go home, my parents always ask me if I have a girlfriend.
			My parents often push me to go on blind dates.
	Social situation	66	I always feel lonely and depressed in social situations, since I have an inferiority complex.
			Given my poor social ability, I am very tired in contemporary social situations.
	…	…	…

**Table 4 ijerph-18-13109-t004:** Keywords indicating social media and examples of sentences.

Sub-Category	Keywords	Count	Examples of Sentences
Consumption of time and energy caused by social media	Mobile phone	235	I can’t sleep without playing the mobile phone.
			I always spend a lot of time with my mobile phone unconsciously, and it even makes me feel sick.
	Video	88	I just want to order a take-out and watch videos after work.
			The video blogs of life often gain a great amount of clicks on video websites. Many people have the desire to have expensive lifestyles as the video blogs show.
	Tik Tok	77	After work I always spend a lot of time in Tik Tok unconsciously.
			I can watch videos on Tik Tok until three o’clock in the morning, even when I have already worked very late.
	…	…	…
Desire for material aroused by social media	Actual life	194	I dream of the high-end lifestyle spreading on social media, but I am unable to make it happen in actual life.
			Our brains are constantly being bombarded with messages on social media, and we begin seeing actual life.
	Friend circle	158	I can see all kinds of colorful life in my WeChat moments.
			I envy the lives of friends shown in my WeChat moments.
	…	…	…
Lives occupied by busy working	Punch a time clock	64	I sometimes wait in the office just to punch a time clock.
			My supervisor thinks that I work hard according to the record of my punching time clock.
	WeChat	59	Even on weekends, I have to work immediately once I receive a message on WeChat.
			Through Wechat, I am often asked to work by my supervisor at very late hours of the night.
	Message	56	I often work all day and forget the time due to various job messages.
			If you are submerged by a mass of messages, you’ll never make anything.
	…	…	…

**Table 5 ijerph-18-13109-t005:** Keywords indicating subjective cognitive bias and examples of sentences.

Sub-Category	Keywords	Count	Examples of Sentences
Insatiable desire	Desire	111	I feel frustrated since I cannot afford my desires.
			My desires cannot be satisfied given my poor ability to pay.
	Satisfy	74	I feel frustrated since my hopes always cannot be satisfied.
			Even my smallest wishes still cannot be satisfied.
	Consumerism	62	We are influenced by consumerism and a money-first culture.
			Some people accept the consumerist and take out a loan.
	…	…	…
The gap between rich and poor	The gap between rich and poor	71	I know that no matter how hard I work, I cannot get my ideal life given the gap between the rich and poor.
			Since the gap between the rich and poor has gradually widened, we are dissatisfied with our current situation.
	Poor	62	Most young people who are born in poverty cannot change their fates even if they do their best.
			Poor kids are still poor when they grow up.
	…	…	…
The gap between dream and reality	Gap	26	I feel frustrated since the huge gap between my dreams and real life.
			The huge gap between my friends’ life and my life makes me feel bitter.
	Frustration	19	I feel frustrated when I recognize that my desires are unrealistic.
			I feel frustrated and anxious when I recognize that my desires always cannot be satisfied.
	…	…	…

**Table 6 ijerph-18-13109-t006:** Comparison with stressors used in the literature.

Stressor	Ours	References	Stressor	Ours	References
Working long hours	Yes	[31,36,42]	Health-care expenses	Yes	[31]
Dead-end job	Yes		Social pressure	Yes	
Rough time during the work	Yes		Consumption of time and energy caused by social media	Yes	[22]
Social life at workplace	Yes	[17]	Desire for material aroused by social media	Yes	
Dislike their jobs	Yes	[17,28,36]	Lives occupied by busy working	Yes	[22]
Commute to work	Yes	[10]	Insatiable desire	Yes	
Lack of social security	Yes		The gap between rich and poor	Yes	
Low salary	Yes	[31,36]	The gap between dream and reality	Yes	
High cost of family life	Yes	[31]	Emotional demands	No	[12,25,36]
High housing costs	Yes	[31,34,35]	Organizational injustice	No	[14,15,36]
High cost to get married	Yes	[31]	Performance stress	No	[8]
Daily consumption	Yes	[31]	Work–family conflict	No	[36]
Education cost	Yes	[31,32]	Being “infected” from other unhealthy organizations	No	[27]
Supporting aging parents	Yes	[30]			

## Data Availability

We collected the raw UGC from “Zhihu”, the largest Chinese social question answering platform. Zhihu, which started in December 2010, provides long-term detailed answers and discusses various questions raised by users. The Zhihu platform is a social question answering platform for people to share their knowledge or opinions. Zhihu users know that their answers are open to the public and the reason why they answer questions is to provide the public with specific knowledge or their views on social issues. There is no acquiescence bias involved as we did not acquire the answers at all.

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
