# Peer review of "Investigating Young Employee Stressors in Contemporary Society Based on User-Generated Contents"

_ijerph, 2021, doi:10.3390/ijerph182413109_

Round 1

Reviewer 1 Report

The article contains cognitively important results of the opinion polls of young Chinese workers, showing their importance on the one hand (the world's leading economy) and the objective specificity of the country's social and economic development problems on the other hand.

STRENGTHS

The article has three strengths:

  1. a) a proposal to use large databases (e.g. social media) to research opinions formed in such a forum
  2. b) the use of linguistic methodology (the frequency of words as the basis for examining the features of individual categories) - which has been used in the areas of HRM research for many years
  3. c) selecting - thanks to the adopted method - three more specific empirically perceivable stressors.

Until now, they have often occurred within the framework of more widely defined stressors, e.g. harassment was an element of interpersonal conflicts (as a result: co-productive behaviors)

Changing the method from questionnaire declarations of respondents to exploring more free statements (question and answer platform) enables an ex post research approach in the assumptions and problems of research (items). It creates new possibilities of identifying and concretizing stressors, depending on the scope of information contained in the analyzed database. Therefore, it enables research based on the resulting information, and not the categories and items created in the preliminary assumptions.

WEAKNESSES

A weakness of the research is also focusing

only on the introduced stressors regarding harassment by superiors, negative external impact and the comparative aspect (constant awareness of lower social status, especially material status). It results directly from the adopted method and research base, i.e. information downloaded from platforms.  This is increased by the mechanisms of Chinese social and economic development that hinder relatively rapid material and family improvement

There is no reference to the above-mentioned Chinese specificity of the development stage with the results of research of other groups of employees. This makes it difficult to assess the generalization of conclusions

The second weakness is methodical - the authors do not rank the introduced stressors in relation to other stressors used in the literature. The mere declarative reference to the JD-R model is too general as a theoretical basis

SUGGESTIONS

It is proposed that the Authors consider the reference to the removal of weaknesses by:

1) a comparison, for example, with the stressors surveyed among Australian workers - with a comparison of similarities and differences

See, e.g., Sawang, S., Newton, C.J. (2018). Defining work stress in young workers. Journal of Employment Counseling · 55: 72-83

2) positioning new stressors in the research achievements to date - see, e.g.

 Law et al. (2020). A systematic review on the efect of work related stressors on mental health of young workers. International Archives of Occupational and Environmental Health. 93: 611–622

In this way, the improvement of the methodological side and conclusions will definitely increase the value of the presented article, which will then be fully prepared for publication

Reviewer 2 Report

I am grateful for the opportunity to review this interesting article. However, it requires some improvements.

In the introduction, the authors should have fundamental articles on stress at work and its classification, e.g.

  • Rout, U. R., & Rout, J. K. (2002). Occupational stress. Stress Management for Primary Health Care Professionals, 25-39.
  • Siegrist, J. (2001). A theory of occupational stress. In J. Dunham (Ed.), Stress in the workplace: Past, present and future(pp. 52–66). Whurr Publishers.

This will help to better organize the theoretical introduction to the article.

Reviewer 3 Report

Thank you very much for the interesting article, which makes an important contribution to the research of occupational demands and resources. The implementation of the human-machine approach as well as the results are described in an understandable and detailed way.

Overall, however, I suggest that the article should be revised.

The human-machine approach is an important step in the analysis of large, existing data sets. Nevertheless, the basis is a qualitative method, namely qualitative content analysis. As an example, the qualitative content analysis according to Mayring is mentioned here (https://www.qualitative-research.net/index.php/fqs/article/view/1089/2385). Please be sure to take these into account in the methods section as well as in the discussion and conclusion.

In addition, I have the following comments:

Lines 43-81: The list of stressors should be summarised in a table for the sake of clarity. This makes it easier for the reader.

Line 89: Please write out abbreviations (JD-R) in full the first time they are mentioned.

Figure 1: Add to signature: Own illustration according to Demerouti and Bakker. In addition, only a section of the JD-R model is shown here. They assume here that resources are only a moderating factor for stress due to occupational demands. However, the JD-R model also shows a direct connection between demands and resources. Please reconsider this and correct it here. Overall, the model should then be presented as a whole and not just an excerpt. The dimension of well-being/motivation is missing here. Self-undermining is only a negative outcome. Please clarify why only this is considered here.

Lines 120-121: The results are already reported here in the brackets. This could be omitted here.

Figure 2: Please revise the flow chart. Hide formatting characters.

Lines 147-154: Questions 1 and 3 are very general and have no direct reference to stressors in the job or to workers. Accordingly, I cannot understand why, for example, question 1 "directly" (line 149) addresses the stressors of young workers. Tiredness is only an outcome here, which can have a variety of causes. What are the considerations behind the choice of questions? This should be explained in more detail. So quite right, all the questions can somehow be related to the job. The selection of the addressed outcomes " tiredness", "desirelessness" and "work overtime" should be justified theoretically.

Line 167: Check number formatting in the text again. here: 3000 - 3,000

Line 168: Please provide details of the evaluation process (as reported in Stage 5): Was a four eyes principle applied? How many staff were involved in the evaluation process? What was the procedure for reaching a decision in the event of ambiguity?

Table 1: Check formatting again, remove unnecessary spaces.

Line 210: "... is not able ..."

Chapter 5: Conclusion: It is important to mention here that the answers stand alone and do not take into account the socio-demographic and socio-economic characteristics of the users or the specific professions. Overall, this is a machine-based qualitative approach to categorisation. The categorisation and interpretation is still required by humans. Future analyses would have to consider the factors in the context of the entire JD-R model and examine direct and indirect effects on and of resources on the stressors.
